**Open Peer Review** | Host-Microbial Interactions | Methods and Protocols

# WormSpot: a machine learning-powered viability scoring platform in *C. elegans* for *Candida* pathogenicity studies

Jonathan Guo Wei Lee,[1] Victor Eng Yong Ong,[1] Dan Zhang[1,2]

**ABSTRACT** Invasive *Candida* infections pose a critical health challenge, exacerbated by emerging antifungal resistance. *Caenorhabditis elegans* (*C. elegans*) offers a genetically tractable and scalable model for studying *Candida* pathogenicity, yet conventional viability assays remain labor-intensive, limiting high-throughput applications. In this study, we developed a machine learning-driven worm viability scoring platform, WormSpot, using representative images of *Candida*-infected worms, and the robust YOLOv8-based framework. By analyzing static morphological features, our model accurately classifies worm viability post-infection, achieving strong concordance with manual scoring, while requiring minimal image input. Validation with well-characterized *Candida albicans* mutants and antifungal agents confirms the platform's robustness in predicting worm survival trends. Notably, WormSpot performs efficiently with diverse image formats and is compatible with standard multi-well microscopy workflows, enabling automated data analysis, and scalable application in various experimental setups. In conclusion, WormSpot provides a data-efficient, reproducible tool for assessing *Candida* virulence using *C. elegans*, supporting both basic pathogenicity studies and antifungal discovery in high-throughput settings.

**IMPORTANCE** Invasive *Candida* infections represent an increasing threat to public health, driven in part by the emergence of antifungal resistance. Although *C. elegans* is a powerful and genetically tractable model for studying fungal virulence *in vivo*, its use has been limited by labor-intensive and poorly scalable viability assays. Here, we present WormSpot, a machine learning-powered platform that enables automated, data-efficient scoring of worm survival from static images. By accurately analyzing high-density worm populations and integrating seamlessly with standard multi-well imaging workflows, WormSpot substantially reduces experimental effort while maintaining robustness and reproducibility. This platform expands the scalability of *C. elegans* infection models for virulence assessment and antifungal screening and provides a generalizable framework for host-pathogen phenotyping in microbiology research.

**KEYWORDS** *Caenorhabditis elegans*, *Candida* infection, machine learning, viability scoring, host-pathogen interactions, automated image analysis, high-throughput screening

Fungal infections present a growing threat to public health, exacerbated by climate change and environmental disturbances. Among these, *Candida* species have become a significant concern due to their high morbidity and mortality rates, especially among vulnerable populations. *Candida albicans* (*C. albicans*) remains the most prevalent etiological cause of candidiasis, while non-*albicans* species such as *C. glabrata*, *C. auris*, and *C. tropicalis* are increasingly recognized, many of which exhibit emerging antifungal resistance, posing significant challenges to therapeutics (1). The dynamic nature of

**Peer Reviewer** Yuan Qiao, Nanyang Technological University, Singapore, Singapore

Address correspondence to Dan Zhang, zhangdan@tll.org.sg.

The authors declare no conflict of interest.

See the funding table on p. 11.

*Candida* pathogenicity and the rapid evolution of multidrug resistance necessitate more efficient and scalable platforms for assessing fungal virulence and antifungal efficacy.

Animal models have long been employed to elucidate host-pathogen interactions in *Candida* research. While mammalian systems, including both whole animals and cell cultures, offer high physiological relevance to human infections and provide valuable mechanistic insights into pathogenesis, their use is often constrained by high costs and limited scalability. The nematode *Caenorhabditis elegans* (*C. elegans*) has been widely utilized as a model organism for studying human diseases and for whole organism-based high-throughput screening (HTS) in drug discovery, owing to its small size, short lifespan, and genetic tractability (2). It has also been established as an alternative animal model for host-pathogen interactions (3), including with *C. albicans* (4), supported by its well-characterized innate immune system that shares conserved signaling pathways with mammals (5). Notably, *C. elegans* can be infected by a range of *Candida* spp., and its transparent body also enables direct visualization of infection progression, making it an attractive host system for studying *Candida* pathogenesis (4, 6, 7), particularly for HTS of mutant libraries or antifungal factors.

*C. elegans* viability assay is commonly used in *Candida* pathogenicity studies and typically based on visual inspection of worm movement or response to stimuli. This conventional approach is labor-intensive and prone to interobserver variability, limiting the scalability and reproducibility required for HTS. While efforts have been made to automate such assays using image analysis tools based on motion detection or static morphological features of worms, these methods have been primarily developed for aging or longevity-related studies, such as WormScan (8), Lifespan Machine (9), WorMotel (10), HeALTH (11), and SiViS (12). Many of these platforms require dedicated physical setups, which restrict their accessibility to research groups with the resources to implement them.

In recent years, machine learning (ML), especially deep learning, has significantly advanced image analysis. Convolutional neural networks (CNNs) have demonstrated enormous potential in identifying and differentiating subtle features within sophisticated image data (13), allowing high-accuracy classification and phenotyping in diverse biological contexts. YOLO (You Only Look Once) ML models (14) have been used to analyze locomotion patterns, predict lifespan, and score morphological traits in *C. elegans* (15–17), reducing the need for dedicated physical setups. However, these approaches have yet to be fully adapted to fungal pathogenesis research, where complex phenotypes and variable morbidity present distinct analytical challenges.

Here, we developed WormSpot, a deep learning-assisted platform for automated worm viability scoring tailored to *C. elegans-Candida* pathogenicity studies. The platform leverages representative post-infection worm images and a custom-trained CNN model based on the YOLOv8 framework (18) to enable automated detection and classification of worm viability after infection. By learning from static morphological features of *Candida*-infected worms, WormSpot shows strong concordance with visual assessments of animal viability.

Unlike movement-based methods, WormSpot relies on morphology-based analysis and can efficiently process high-density worm images that exceed practical manual limits. This capability reduces experimental scale and image data volume, minimizes subjectivity in manual scoring, increases assay throughput, and lowers computational demands, making the platform highly suited for HTS applications. Moreover, it eliminates the need for specialized hardware or large data storage, as viability assessments are performed on single-frame images.

We validated WormSpot using well-characterized *C. albicans* mutants with defined virulence profiles, as well as drug interventions involving established antifungal agents. Additionally, we demonstrated a screening application employing this platform, facilitating the identification of potential novel antifungal factors with markedly accelerated data analysis. Importantly, WormSpot operates efficiently across diverse image formats and can be readily adapted to various experimental contexts, including

standard multi-well setups. Collectively, our work establishes a scalable and data-efficient approach for assessing *Candida* pathogenicity using *C. elegans*, offering a robust and versatile tool for both basic research and high-throughput applications.

## RESULTS

### Characteristic worm morphologies under *Candida* infection offer a reliable visual basis for image-based viability scoring

In liquid medium, live worms typically exhibit a sinusoidal shape, while dead worms appear stiff and straightened (Fig. 1A). These morphological features are commonly used as visual cues for scoring worm viability (19). Following *C. albicans* infection, worms killed by the fungus often display hyphal protrusions emerging from the body (20) (Fig. 1A), providing an additional indicator of mortality. Such characteristic worm morphologies hold promise for viability scoring in *C. albicans* infection assays using images.

To evaluate the effectiveness of an image-based viability scoring method, we imaged infected worms in 96-well plates daily over four days. Each day, eight frames from the same well were captured at ~49 s intervals. Videos of 351 selected worms were analyzed to determine viability based on mobility (Fig. S1A), defined as "Ground truth" in Fig. 1B. The same worms were independently classified as "Alive" or "Dead" based on the first frame and the morphological features shown in Fig. 1A, referred to as "Eye prediction" in Fig. 1B. This image-based classification achieved high accuracy: 92.6% (worm number [m] = 176) for identifying live worms and 83.4% (m = 175) for dead worms. Misclassified dead worms typically retained a sinusoidal shape but showed no movement across frames, whereas misclassified live worms were often captured in a transient straight posture (indicated by arrows in Fig. S1A). These results suggest that worm shape-based viability scoring from images is reliable and applicable to *Candida* infection assays.

### Establishment of WormSpot: A YOLOv8-based viability scoring platform

Given the reliability of image-based classification, we next developed WormSpot, a deep learning-driven viability scoring platform tailored to worm-*Candida* infection studies, using representative images of infected worms and the YOLOv8 algorithm (18) (Fig. 2A; see Materials and Methods for details). Our model aims to replace conventional labor-intensive viability inspection with automated, high-accuracy image-based scoring suitable for both routine analyses and high-throughput applications. We also generated diverse data sets incorporating both worm-level and well-level assessments to evaluate the robustness and applicability of WormSpot in worm classification (Fig. 2A).

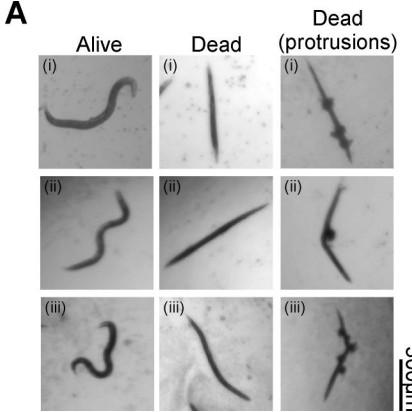
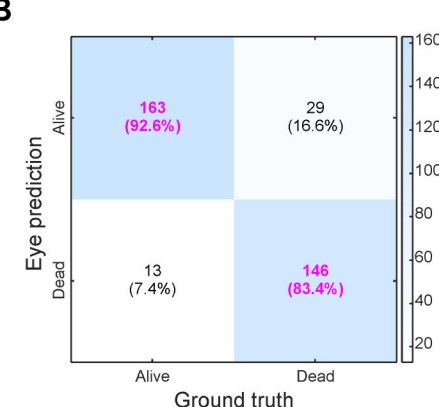

**FIG 1** Characteristic morphologies of *Candida*-infected worms offer a reliable visual basis for viability scoring. (A) Example worm images with indicated status under *C. albicans* infection. Scale bar, 300 µm. (B) Confusion matrix showing correct and incorrect predictions of the eye-annotated status for 176 live and 175 dead worms. Correctly predicted fractions are highlighted in pink.

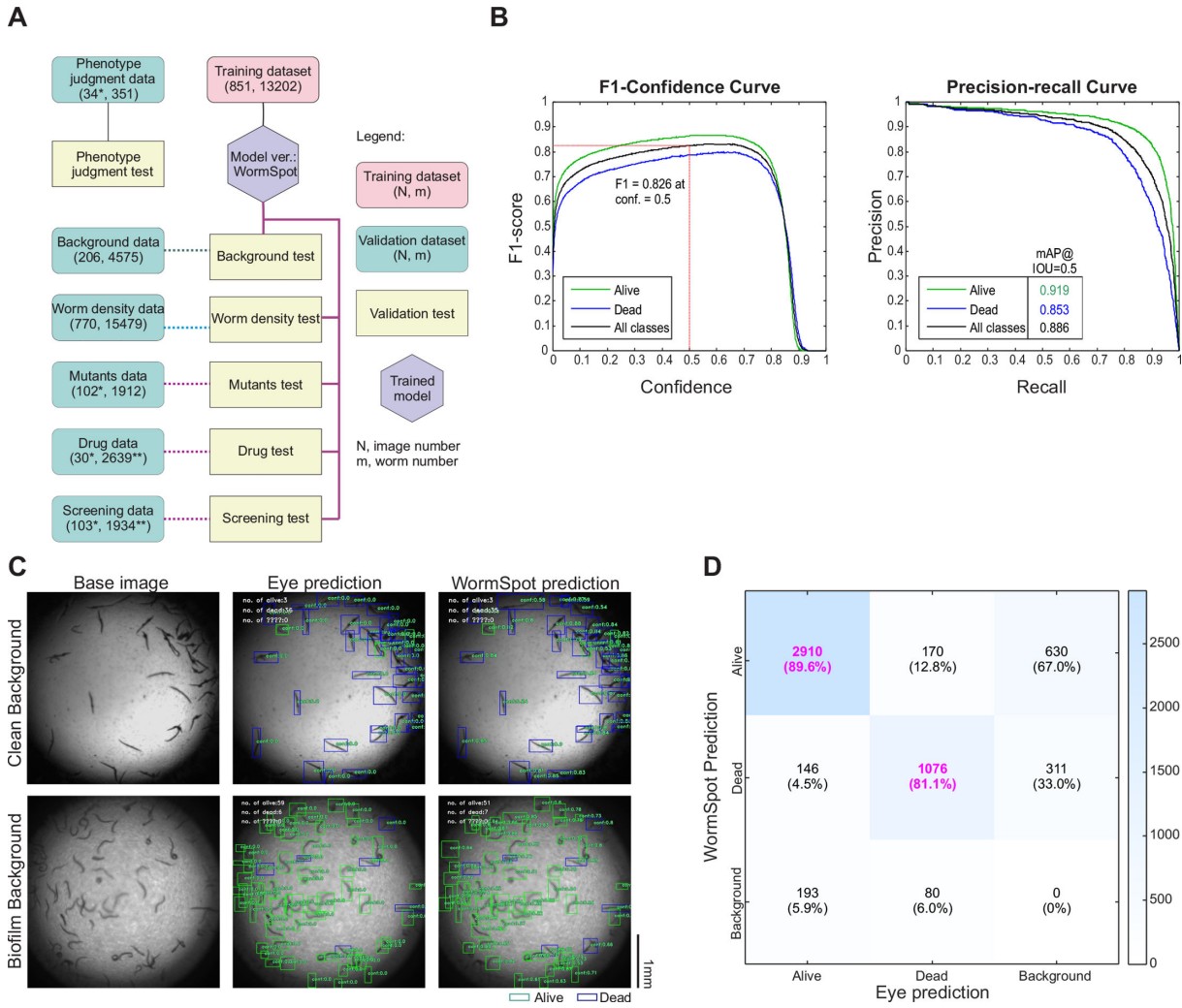

**FIG 2** WormSpot offers reliable worm-based viability scoring. (A) Infographic illustrating the training and testing workflow for establishing and validating WormSpot. N*, total number of images at the first time point; m**, total number of worms detected by WormSpot at the first time point. Otherwise, N denotes the total number of raw images, and m shows the total number of eye-annotated worms. (B) Validation metrics of WormSpot on the background test data set, including F1 scores, precision-recall curve, and mean average precision (mAP) at 0.5 intersection over union (IoU). The red dotted line indicates the F1 score for "all classes" at a 0.5 confidence threshold. (C) Example clean and biofilm-dense background images with corresponding eye annotations and WormSpot predictions are shown. Scale bar, 1 mm. (D) Confusion matrix comparing WormSpot predictions with eye annotations at IoU and confidence thresholds of 0.5, with correct predictions highlighted in pink. "Background object" refers to a worm missed by one prediction source but detected by the other. Worm counts and corresponding fractions of each eye-annotated category (i.e., "Alive," "Dead," and "Background object") are included.

Typically, deep learning algorithms are evaluated using precision and recall. Precision indicates how many predicted positives are correct, while recall reflects how many actual positives are identified. The F1 score, defined as the harmonic mean of precision and recall, provides a combined measure of model performance. WormSpot achieved high F1 scores for "Alive" and "Dead" classifications both individually and overall (Fig. 2B), suggesting high model accuracy.

Notably, a major challenge in our image-based worm detection and classification is background variability caused by *C. albicans* biofilm formation in the wells during the late stages of infection. Our model performed well on 103 test images with clean backgrounds and was slightly less accurate on another 103 test images with relatively biofilm-dense backgrounds (Fig. 2C). Overall, a high proportion of "Alive" and "Dead" classifications were correctly predicted at the individual worm level, with a total of 4,575 eye-annotated worms (Fig. 2D, highlighted in red).

## Assessing WormSpot effectiveness and applicability for well-based predictions

For high-throughput well-based applications, an optimal model is expected to deliver accurate predictions from a single image representing a well. We first assessed both the total worm numbers as well as counts for each class (i.e., "Total," "Alive," and "Dead") using the same set of 206 test images with varying backgrounds. For each category, a well was considered a "pass case" if the predicted counts were within a specified ± threshold percentage of the manual counts. For wells with low worm numbers, the tolerance was set to at least ±1 worm.

WormSpot demonstrated overall reliable performance, achieving passing rates of 83.9%, 76.2%, and 73.3% for "Total," "Alive," and "Dead" worm counts at a 15% threshold, which further improved to 92.2%, 86.4%, and 75.2%, respectively, at a 20% threshold (Fig. 3A). The higher accuracy in total worm counts indicates that our model is more effective at worm detection than classification. When analyzing images with clean and biofilm-dense backgrounds separately, we found that the model performed particularly well on clean-background images, with high passing rates across all three categories (Fig. S2A). However, predictions for "Dead" worms in biofilm-dense background images were less accurate, with ~60% passing rates at both thresholds (Fig. S2A).

Although predicted counts reflect model performance, survival rate remains the primary readout in worm viability assays for *Candida* pathogenicity studies. In principle, predicted survival rate can remain accurate despite some imprecision in individual worm counts. We thus proceeded to evaluate survival rate prediction using our model. Moreover, reliable estimation of survival trends at the single-well level requires a sufficiently large worm population, which a robust model must effectively accommodate.

To this end, we assembled a test data set comprising 770 images with varying worm densities per well (Fig. 2A). WormSpot exhibited high accuracy in predicting survival rates, with over 90% of wells passing at 15% and 20% tolerance thresholds (Fig. 3B). Notably, the "10–30" and "30–60" worms-per-well groups consistently showed high passing rates across all thresholds (Fig. 3B), suggesting these as optimal worm density ranges for model application. In contrast, survival rate predictions were less accurate for wells with low worm density ("<10" worms per well), as small fluctuations in worm counts lead to disproportionately large deviations in the calculated survival

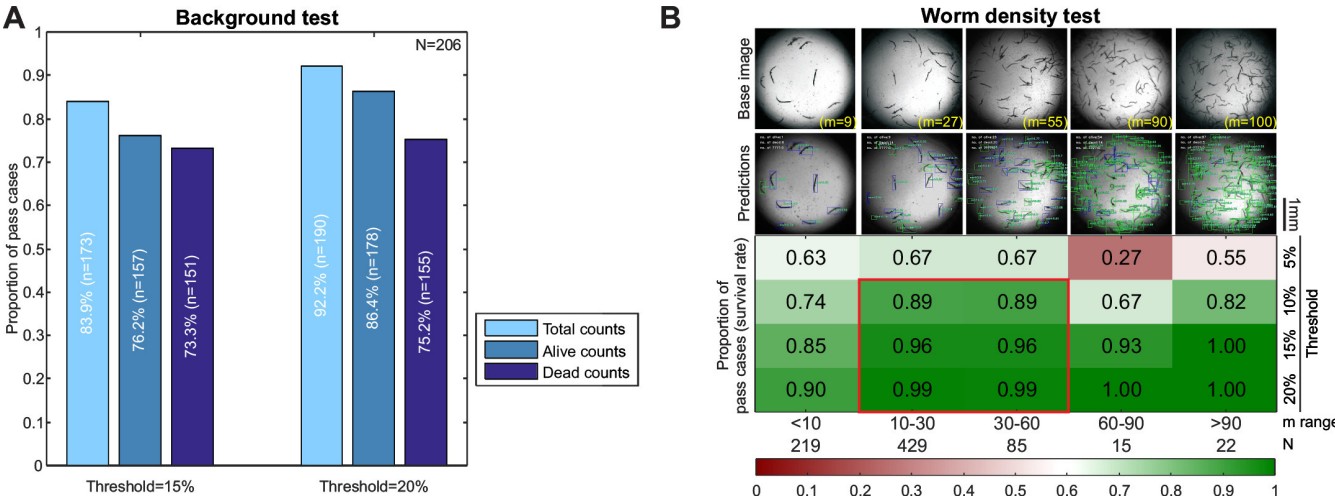

**FIG 3** WormSpot offers reliable well-based worm viability scoring. (A) Bar plot showing fractions of pass cases in the background test at 15% and 20% tolerance thresholds for the indicated counts. N, total number of images; n, number of pass cases. (B) Heatmap showing fractions of pass cases for survival rate across five indicated m ranges in the worm density test under four tolerance thresholds (5%, 10%, 15%, and 20%). Shown are representative images from each worm density category, along with corresponding WormSpot predictions and eye-annotated total worm counts. m, eye-annotated worm count per well; N, image number per m range. Scale bar, 1 mm.

ratio. Consistent with the background test results, more fail cases for "Dead" counts were seen across all worm density ranges (Fig. S2B). Importantly, despite underperformance in count predictions within the "10–60" worms-per-well range at 10%–20% thresholds (Fig. S2C), survival rate estimates remained robust (Fig. 3B), highlighting the model's reliability in capturing overall viability trends even when absolute count predictions were less precise.

## Application of WormSpot in assessing *Candida* pathogenicity

Given WormSpot's robustness in well-based survival rate prediction, we next applied it to assess *Candida* pathogenicity using known *C. albicans* mutants and antifungal treatments in infection assays. Virulent wild-type strains SC5314 and BWP17UH, along with the avirulent heat-killed SC5314, were included as positive and negative controls, respectively. We also tested the low-virulence auxotrophic strain BWP17, the non-invasive *hgc1Δ/Δ* mutant defective in hyphal formation (21), and the *bcr1Δ/Δ* mutant, which is impaired in biofilm formation and exhibits reduced virulence (22). Remarkably, WormSpot accurately recapitulated survival trends from manual scoring over three days post-infection, consistent with previously reported pathogenicity profiles (21, 22) (Fig. 4A). Likewise, it clearly captured the expected improvement in survival of

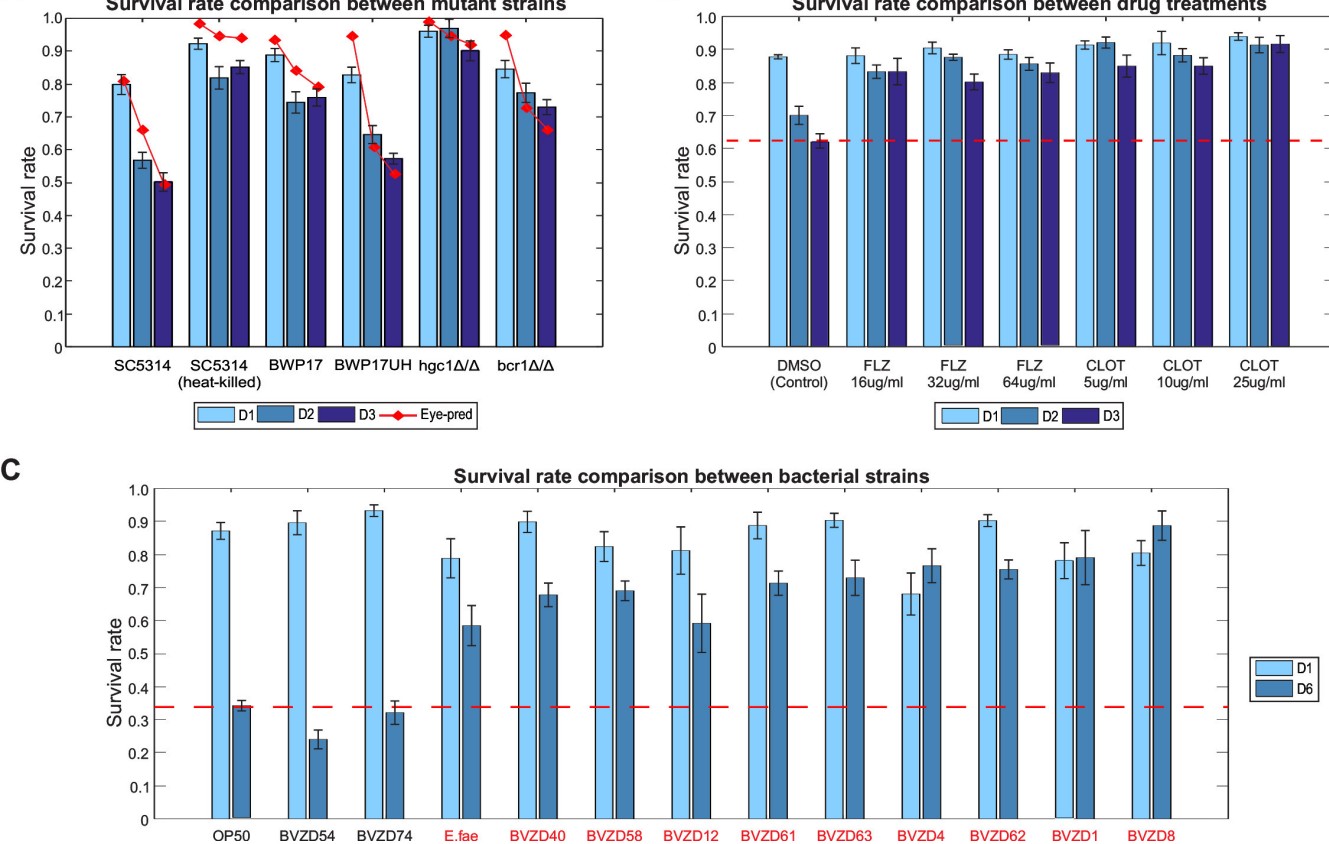

**FIG 4** WormSpot enables accurate viability-based assessment of *Candida* pathogenicity in *C. elegans*. (A) Bar charts showing the survival rates of worms infected with indicated *C. albicans* strains over the first three days post-infection (D1–D3; worm number (M) > 200 for each mutant). The red line indicates survival rates determined by manual scoring. (B) Bar charts showing the survival rates of worms infected with wild-type *C. albicans* SC5314 and treated with indicated drugs over the first three days post-infection (m > 300 for each condition). The red dotted line marks the survival rate of the DMSO control group on D3 for comparison. (C) Bar charts showing survival rates of SC5314-infected worms treated with indicated bacteria isolates from an example screening on D1 and D6 post-infection (m > 70 for each isolate tested). The red dotted line marks the survival rate of the OP50 control on D6 for comparison. Isolates prolonging worm survival are highlighted in red. All analyses were performed with at least four biological replicates showing consistent trends. Error bars in (A–C) represent standard deviations across all replicates.

SC5314-infected worms following treatment with either of the widely used antifungal drugs fluconazole (FLZ) or clotrimazole (CLOT) (Fig. 4B).

To further evaluate WormSpot's capacity for high-throughput applications, we conducted a well-based screen of 155 bacterial isolates from our private lab collection to identify those potentially effective against *C. albicans* infection in worms. Bacteria were introduced to worms one day prior to infection and maintained throughout the infection period. To maximize data efficiency, each well was imaged twice, on day one and day six post-infection. After a few rounds of screenings, we identified 10 bacterial isolates that significantly prolonged the survival of SC5314-infected worms (Fig. 4C), including *Enterococcus faecalis* (*E. fae*), which was previously shown to reduce *C. albicans* virulence in worms (23–25). This screening example clearly demonstrates the strong potential of our platform for scalable applications and its ability to facilitate the discovery of novel antifungal factors through efficient data analysis.

## WormSpot applicability to general worm images

Building on the demonstrated strength of WormSpot in high-throughput applications, we next explored its applicability to diverse worm images beyond our original training format (~1,000 × 1,000 pixels, 3 µm/pixel). We employed three publicly available image data sets that differ in resolution and background texture and compared WormSpot's performance with two established YOLO-based models—Generalist (16) and Deep-Worm-Tracker (15)—for worm detection. Notably, these two models do not classify worm viability.

Among them, the benchmark BBBC010 contains worm images acquired from 384-well plates (26); CSB-1 features images captured from agar plates (27); and Synthetic (28) includes images generated by randomly modifying CSB-1. WormSpot accurately detected 91.1%, 83.6%, and 89.7% of worms from these data sets, respectively (total worm counts: m = 1,419 for BBBC010; 60,571 for CSB-1; and 174,898 for Synthetic), consistently outperforming Generalist and Deep-Worm-Tracker (Fig. S3A). We also compared the well-based worm counting performance of the three models using our worm density testing data set ($N$ = 770 images) at a 15% tolerance threshold. Remarkably, while WormSpot achieved a 79.6% passing rate, both Generalist and Deep-Worm-Tracker performed poorly, at passing rates of only 3.1% and 17.4%, respectively (Fig. S3A). Collectively, these results highlight WormSpot's adaptability to diverse worm images from various experimental setups and further demonstrate its robustness in worm recognition.

## DISCUSSION

With the rising threat of emerging *Candida* pathogens to public health, there is an urgent need for efficient and scalable tools to assess *Candida* virulence, elucidate pathogenesis, and develop effective therapies. To address this need, *C. elegans* has emerged as a powerful whole-organism model that enables high-throughput studies for both mechanistic investigations and antifungal drug discovery. Long before the revolutionary application of deep learning to image detection and classification, significant efforts were dedicated to advancing automated imaging and analytical tools for worm-based high-throughput assays. For instance, WorMachine employed MATLAB-based frameworks to extract morphological features, quantify fluorescence, apply ML techniques for complex worm image analyses (29). Other platforms for worm survival assays, such as the Lifespan Machine, WorMotel, and SiViS, often require specialized experimental setups and computational infrastructure (9, 10, 12).

As a breakthrough in object detection, YOLO-based deep learning frameworks offer an exceptional balance of speed, accuracy, and computational efficiency compared to traditional two-stage detectors. However, recent worm-specific YOLO models (e.g., Deep-Worm-Tracker [15] and Generalist [16]) lack viability assessment functionality and perform poorly even in basic worm detection during *Candida* infection, likely due to the complexity of worm phenotypes and image background textures. This highlights the

need for a dedicated model for worm survival assays, which are routinely used in *Candida* pathogenicity studies.

Our tailored WormSpot effectively addresses this gap by achieving reliable accuracy in both worm-level and well-level viability scoring, making it highly suitable for high-throughput applications. WormSpot was trained on manual viability annotations guided by visual assessment of morphological features, which itself achieved ~92.6% accuracy for "Alive" and 83.4% for "Dead" worms, providing a reliable foundation. The high annotation accuracy, supported by clearly discernible morphological features, indicates that subjectivity inherent to manual scoring is minimal. The resulting model attains an overall strong F1 score of 0.826 at a 0.5 confidence threshold and a high mean average precision (mAP)@0.5 score of 0.886, reflecting high model accuracy. It also performs efficiently across three public worm image data sets and consistently outperforms Deep-Worm-Tracker and Generalist in worm detection, particularly on our image data set. This underscores our model's high adaptability for analyzing worm images from various experimental setups, including Petri dishes, 384-well plates, and 96-well plates.

However, accurate identification of dead worms in biofilm-dense background images and the processing of overcrowded worm images remain challenging for WormSpot. Dead worms are often obscured by fungal biomass and progressively lose distinct morphological features over time, making reliable detection difficult. Despite these limitations, survival rate predictions remain robust, with deviations generally within 10%–20% for wells containing 10–60 worms, a range considered ideal for single well-based survival assays. Importantly, WormSpot accurately captures early survival trends in worm infection assays using *C. albicans* mutants and drug interventions, yielding results consistent with both manual scoring and established pathogenicity profiles. Our screening example further demonstrates its strong potential for scalable applications. Notably, the platform can process ~160 images per minute using average computational resources, whereas manual scoring typically requires several minutes per image, representing a substantial improvement in analytical efficiency. Its compatibility with standard multi-well microscopy workflows also allows broad applicability across diverse experimental contexts.

In summary, we developed WormSpot as a reliable viability scoring platform tailored for assessing *Candida* pathogenicity using *C. elegans*, featuring automated scalability to support both routine analyses and large-scale screening. Given that *C. elegans* exhibits similar morphological and viability changes upon infection with a wide range of other microbial pathogens, including bacteria, fungi, and viruses (30, 31), WormSpot has strong potential for extension to virulence analyses beyond *Candida*.

## MATERIALS AND METHODS

### Experimental model and study participant details

#### C. elegans and C. albicans strains

The *C. elegans* strain AU37 (*glp-4*(*bn2*) I; *sek-1*(*km4*) X) (19) was obtained from the Caenorhabditis Genetics Center. *C. albicans* strains SC5314, BWP17 (*ura3–/–*; *his1–/–*; *arg4–/–*), BWP17UH (*ura3+/+*; *his1+/+*; *arg4–/–*), *hgc1Δ/Δ* (*hgc1Δ::ura3+/Δ::his1+*), and *bcr1Δ/Δ* (*bcr1Δ::ura3+/Δ::his1+*) were kindly provided by Dr. Wang Yue's lab.

### Method details

#### C. elegans infection assay

Worm infection assays were conducted as previously described (25) with minor modifications. Briefly, synchronized adult AU37 worms were exposed to freshly cultured *C. albicans* strains for 6 h at 24°C. Following infection, worms were thoroughly washed and transferred to 96-well plates, which were vortexed prior to imaging over multiple days. Typically, images used for both model training and testing were manually

annotated and scored, while those used for testing were also independently analyzed by WormSpot to classify worms as "Alive" or "Dead." Survival rates were calculated accordingly.

For drug treatment experiments, synchronized adult AU37 worms were infected with *C. albicans* strain SC5314 in the presence of fluconazole (Sigma-Aldrich, F8929-100MG) or clotrimazole (Sigma-Aldrich, C6019-5G) at the indicated concentrations for 6 h. Drugs were maintained post-infection. A 25 µL/mL dimethyl sulfoxide (DMSO) solution was used as the control. Scanned images were analyzed by WormSpot, and survival rates were calculated accordingly.

For bacterial screening, synchronized L4 AU37 worms were pre-exposed overnight to washed, stationary-phase bacterial cultures prior to infection with *C. albicans* strain SC5314. The respective bacterial strains were present during infection, and their trace amounts were retained post-infection. Scanned images were analyzed by WormSpot and survival rates were calculated accordingly. Figure 4C presents the data for the laboratory feed OP50 control, 10 bacterial isolates that significantly prolonged survival of infected worms, and two randomly selected isolates.

## Microscopy

Imaging was performed using an upright Leica DM6000B microscope equipped with a Leica HC FL PLAN 2.5×/0.07 HC objective lens, a DFC7000 GT camera (Leica, Germany), and an automated SCAN stage system 100 × 100 (Märzhäuser Wetzlar), controlled by Leica Application Suite X. Typically, each well of the 96-well plate was manually focused and imaged once at a single focal plane for each time point.

## Quantification and statistical analysis

### Model training for WormSpot

#### Training and detection environment

All computational tasks were performed on a workstation equipped with an Intel(R) Xeon(R) W-2125 CPU (4.00 GHz), NVIDIA GeForce GTX 1080 GPU (8GB VRAM), 128 GB RAM, a 500 GB SSD (Samsung MZVLB1T0HALR-000L7), running Windows 10 (Version 22H2, OS build 19045.5965). The YOLOv8 architecture was selected for model training. All scripts were written and tested in Python 3.11.4 using Jupyter Notebook 6.5.4. Model training was performed with the !yolo command-line interface from Ultralytics (version 8.3.149), with parameters set to epochs = 100 and imgsz = 640. A total of 2,553 images, including augmented data, were used for training.

#### Training data set preparation

We selected 851 images of individual wells containing infected worms with varying densities and background quality. As illustrated in Fig. S3B, each original image (~1,920 × 1,440 pixels, 3 µm/pixel) was processed in ImageJ using the built-in "Mean Dark" and "Analyze Particles…" functions to define the well ROI and crop the image to ~1,000 × 1,000 pixels. These cropped images were then used for annotation and training.

Out-of-focus images with blurry subjects or overexposed images are common causes of poor predictions. To address this, two additional images were generated for each cropped image using Roboflow's augmentation feature at https://roboflow.com/ and added to the training set. These augmented images had randomized exposure adjustments (±25%) and blur applied (up to 7.6 px). The final training data set consisted of 851 cropped images and 1,702 augmented images.

#### Data annotation

Training images were manually annotated using Roboflow's web-based tool, with bounding boxes denoting worms as "Alive" or "Dead." Preliminary YOLO models trained on the parts of the data set generated predictions on new images, which were then

manually reviewed. This annotation and prediction process was iteratively repeated until all images were labeled.

## Worm-based prediction validation

The test images were similarly cropped as above and manually annotated using Roboflow as "Eye prediction" for comparison and validation. Corresponding WormSpot predictions were evaluated using the YOLO validation command (Ultralytics, https://docs.ultralytics.com/modes/val/) with a confidence threshold of 0.5 and an intersection over union (IoU) of 0.5, returning the following metrics: (1) confusion matrix, (2) F1 curve, (3) precision curve, (4) recall curve, and (5) precision-recall curve. This worm-based validation was performed only on the background test shown in Fig. 2.

## Well-based prediction analysis

Similar to worm-based validation, the test data set was manually annotated as "Eye prediction" and independently predicted by WormSpot (32). For each test image, worm counts ("Total," "Alive," "Dead") and survival rate (i.e., "Alive" count divided by "Total" count) from "Eye prediction" were compared to the predicted values. A "pass"case was defined if the predicted values fell within a specified tolerance threshold (e.g., 5%, 10%, 15%, 20%) of the manual scores. To avoid bias against wells with low worm counts, the tolerance threshold was set with a minimum value of 1, since higher counts inherently permit larger numerical differences. This analysis was performed for the background test (Fig. 3A; Fig. S2A) and the worm density test (Fig. 3; Fig. S2B and C). Group survival rates shown in Fig. 4 were obtained by averaging the survival rate from each replicate (well), with standard deviations across replicates shown as error bars.

## Model comparison

We compared the worm detection performance of WormSpot to Generalist (based on YOLOv8n) and Deep-Worm-Tracker (based on YOLOv5) using the benchmark BBBC010, CSB-1, and Synthetic data sets described previously (26, 28), as well as the worm density test data set established in this work. The BBBC010 data set contains 100 well images of live and dead worms from a 384-well plate, while the CSB-1 and Synthetic data set consist of 4,631 and 10,000 images from agar plates, respectively. We validated the three models using the !yolo command line to obtain confusion matrix scores on the BBBC010, CSB-1, and Synthetic data sets. The $m_{correct}$ was defined as the percentage of worms correctly detected.

In addition, we tested all three models on our worm density data set to predict total worm counts at a tolerance threshold of 15%. The $N_{correct}$ calculates the percentage of "pass" cases as defined above. Notably, we did not evaluate the "Alive" and "Dead" statuses in this test, as the other two models do not support viability assessment.

## ACKNOWLEDGMENTS

We greatly thank the Temasek Life Sciences Laboratory for funding support. We thank Prof. Yue Wang for sharing strains and reagents used in this work, and are grateful to Ian Cheong, Huile Lim, Zixin Wong, Guisheng Zeng, and Eve Wai Ling Chow for their technical assistance. We also thank all Zhang lab members for discussions and suggestions throughout the study.

D.Z. and J.G.W.L. conceived the project. J.G.W.L. and V.E.Y.O. trained the model and performed analyses. V.E.Y.O. performed worm infection assays. J.G.W.L. and D.Z. drafted the paper. V.E.Y.O. contributed to the manuscript.

## AUTHOR AFFILIATIONS

[1]Temasek Life Sciences Laboratory, National University of Singapore, Singapore, Singapore

²Department of Biological Sciences, National University of Singapore, Singapore, Singapore

## AUTHOR ORCIDs

Dan Zhang ⓘ http://orcid.org/0000-0002-9532-2090

## FUNDING

| Funder | Grant(s) | Author(s) |
| --- | --- | --- |
| Temasek Life Sciences Laboratory | | Jonathan Guo Wei Lee |
| | | Victor Eng Yong Ong |
| | | Dan Zhang |

## AUTHOR CONTRIBUTIONS

Jonathan Guo Wei Lee, Conceptualization, Data curation, Formal analysis, Investigation, Methodology, Software, Validation, Visualization, Writing – original draft, Writing – review and editing | Victor Eng Yong Ong, Data curation, Formal analysis, Investigation, Methodology, Validation, Visualization, Writing – review and editing | Dan Zhang, Conceptualization, Funding acquisition, Methodology, Project administration, Resources, Supervision, Writing – original draft, Writing – review and editing, Investigation

## DATA AVAILABILITY

Further information and requests for resources and reagents should be directed to and will be fulfilled by the lead contact, Dan Zhang (zhangdan@tll.org.sg). Codes generated in this paper are available at https://github.com/dbszdan/WormSpot.

## ADDITIONAL FILES

The following material is available online.

### Supplemental Material

**Supplemental figures (Spectrum00252-26-s0001.docx).** Fig. S1 to S3.

### Open Peer Review

**PEER REVIEW HISTORY (review-history.pdf).** An accounting of the reviewer comments and feedback.

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
