## [Reviewer comments · Microbiology Spectrum]

Microbiology Spectrum

WormSpot: a machine learning-powered viability scoring platform in *C. elegans* for *Candida* pathogenicity studies

Jonathan Lee, Victor Ong, and Dan Zhang

Corresponding Author(s): Dan Zhang, Temasek Life Sciences Laboratory

Review Timeline:

Submission Date:	January 20, 2026
Editorial Decision:	February 22, 2026
Revision Received:	March 11, 2026
Accepted:	March 24, 2026

Editor: Luis Martinez

Reviewer(s): Disclosure of reviewer identity is with reference to reviewer comments included in decision letter(s). The following individuals involved in review of your submission have agreed to reveal their identity: Yuan Qiao (Reviewer #1)

Transaction Report:

DOI: <https://doi.org/10.1128/spectrum.00252-26>

Re: Spectrum00252-26 (**WormSpot: a machine learning-powered viability scoring platform in *C. elegans* for *Candida* pathogenicity studies**)

Dear Dr. Dan Zhang:

Thank you for the privilege of reviewing your work. Below you will find my comments, instructions from the Spectrum editorial office, and the reviewer comments.

Revision Guidelines

Sincerely,
Luis Martinez
Editor
Microbiology Spectrum

Reviewer #1 (Comments for the Author):

The manuscript is well written and clear to read. The work presented in sound and offers a valuable analytical tool to the field. My only question is regarding appropriate statistical analysis needed for all the bar graphs and error bars are missing.

In general, the survival rates of worm are well-above 0.5 in most of the data represented, the threshold of the worm-infection

assay is quite narrow, a wider window of survival versus killing would be desirable.

Reviewer #2 (Comments for the Author):

This manuscript presents WormSpot, a novel deep learning background for automated quantification of *C. elegans* survival in *Candida* infection assays. The authors show that morphology based classification can accurately estimate survival in wells with dense biofilm and high fungal growth, where other motion based approaches fail. The study addresses a critical limitation in antifungal discovery assays and provides a solid validation across multiple fungal species and antifungal treatments. Addressing a few points related to framing, ground truth definition, and interpretations would further strengthen the manuscript.

Comments:

1. While the technical performance of WormSpot is described, its advantages over existing *C. elegans* analysis tools could be highlighted more clearly. Most current platforms rely on motion detection. In contrast, WormSpot emphasizes single frame, morphology based viability assessment under infection, which is particularly relevant for fungal biofilms. Therefore, the authors should clearly distinguish WormSpot from other movement-based survival scoring systems.
2. The manuscript refers to "ground truth" based on eye-based scoring, which may raise concerns about subjectivity. Therefore, the authors should clearly describe how ground truth viability was defined.
3. The authors mentioned that dead worms are frequently misclassified in dense biofilm backgrounds. They should add that dead worms lose clear morphological features and are covered by fungal biomass, while overall survival estimates remain reliable despite some misclassification.
4. Replace "dirty background" with "biofilm background".

Reviewer 1

We thank the reviewer for the suggestions to improve this paper.

The reviewer has the following points.

The manuscript is well written and clear to read. The work presented is sound and offers a valuable analytical tool to the field. My only question is regarding appropriate statistical analysis needed for all the bar graphs and error bars are missing.

Thank you for your comment. As suggested, we have updated all bar charts in Figure 4 to display average survival rates with error bars (standard deviations across all replicates). The statistical methods have also been added to the Methods section of the revised manuscript (Page(p). 19)

In general, the survival rates of worm are well-above 0.5 in most of the data represented, the threshold of the worm-infection assay is quite narrow, a wider window of survival versus killing would be desirable.

Thank you for the comment.

We assume the reviewer referred to the survival rates shown in Figures 4A and 4B. We understand the point raised, as functional studies often display a wider range of survival versus killing in viability assays. In our study, we deliberately focused on early infection data, which inherently has a narrower survival-killing window, to demonstrate that WormSpot can detect trend differences at early stages - differences that are often not apparent to the naked eye. This feature is particularly valuable for fast-paced, high-throughput screening experiments with shorter infection periods. We hope that highlighting the earliest point at which our model can capture differences will be of interest to readers seeking methods for early detection.

Reviewer 2

We thank the reviewer for the critical comments and for the suggestions to improve this paper.

While the technical performance of WormSpot is described, its advantages over existing *C. elegans* analysis tools could be highlighted more clearly. Most current platforms rely on motion detection. In contrast, WormSpot emphasizes single frame, morphology based viability assessment under infection, which is particularly relevant for fungal biofilms. Therefore, the authors should clearly distinguish WormSpot from other movement-based survival scoring systems.

Thank you for the suggestion. We now included a discussion of the limitations of existing systems (p.5), and have clearly highlighted the advantage of WormSpot, as suggested, in the introduction (pp. 5-6) - “Unlike movement-based methods, WormSpot relies on morphology-based analysis and can efficiently process high-density worm images that exceed practical manual limits. This capability reduces experimental scale and image data volume, minimizes subjectivity in manual scoring, increases assay throughput, and lowers computational demands, making the platform highly suited for HTS applications. Moreover, it eliminates the need for specialized hardware or large data storage, as viability assessment are performed on single-frame images.”

The manuscript refers to "ground truth" based on eye-based scoring, which may raise concerns about subjectivity. Therefore, the authors should clearly describe how ground truth viability was defined.

First, to clarify, the ‘ground truth’ shown in Figure 1B was determined based on worm movement in videos (sample time-lapse frames are shown in Figure S1), which has been explicitly stated in both the original (p. 6) and revised manuscript (p. 7).

We performed an initial comparison between single-frame eye-based predictions (using the first frame of each video) and this defined ‘ground truth’ to demonstrate that single-frame, morphology-based prediction is reliable, achieving 92.6% and 83.4% prediction accuracy for live and dead worms, respectively (Figure 1B). Therefore, we believe that the subjectivity inherent to eye-based scoring is minimal. This also provided a robust foundation for training WormSpot.

That said, we have included a related discussion (p. 11) as well as the limitations of WormSpot (p. 12), which may partly arise from the subjectivity and visual limitations associated with the training method.

The authors mentioned that dead worms are frequently misclassified in dense biofilm backgrounds. They should add that dead worms lose clear morphological features and are covered by fungal biomass, while overall survival estimates remain reliable despite some misclassification.

Thank you for the suggestion.

We now included a suggested sentence to the discussion – “Dead worms are often obscured by fungal biomass and progressively lose distinct morphological features over time, making reliable detection difficult. Despite these limitations, survival rate predictions remain robust, ...” on p. 13.

Replace "dirty background" with "biofilm background".

As suggested, we replaced all “dirty background” with “biofilm-dense background” in the test and with “biofilm background” in the figures (Figures 2C and S2A).

Re: Spectrum00252-26R1 (**WormSpot: a machine learning-powered viability scoring platform in *C. elegans* for *Candida* pathogenicity studies**)

Dear Dr. Dan Zhang:

Your manuscript has been accepted, and I am forwarding it to the ASM production staff for publication. Your paper will first be checked to make sure all elements meet the technical requirements. ASM staff will contact you if anything needs to be revised before copyediting and production can begin. Otherwise, you will be notified when your proofs are ready to be viewed.

Sincerely,
Luis Martinez
Editor
Microbiology Spectrum